

# Use of soybean as an alternative protein source for welfare-orientated production of American alligators (*Alligator mississippiensis*)

Mark Flint and Jaylene Flint

One Welfare and Sustainability Center, College of Veterinary Medicine, The Ohio State University, Columbus, Ohio, United States

## ABSTRACT

Soybean meal based diets have been suggested to cause gastrointestinal issues in certain species when used as a protein alternative. Using a randomized design, we tested 1,728 alligators in one of 13 communal pens offered one of two diets (seven pens ($n = 928$) of soybean-based protein diets and six pens ($n = 800$) of animal-based protein diets) to determine if soybeans would negatively affect the growth, hide quality, behavior and health, when compared with an otherwise identical diet using animal-based protein. Both feeds were nearly identical in composition for protein and fat percentages and identical for all minerals and elements. Crude protein was a minimum of 50%, crude fat a minimum of 12%, crude fiber a minimum of 4%, and phosphorous was maintained at 1%. From this information we estimated the One Welfare of using soy as a protein in commercial diets. Although there was a statistically significant decrease in belly width ($p = 0.0009$; harvested hide size) for alligators fed soybean-based protein diets, all other measured parameters of soybean *vs.* animal-based protein diets were comparable, suggesting this environmentally sustainable alternative protein source warrants consideration as a feed base. Weight was not significantly different suggesting either diet would yield similar volumes of meat. Total length was significantly affected by diet. Hide quality was not negatively impacted by protein type, with both diets producing high quality hides free of defects (assessed at the salted hide stage prior to tanning). Behaviors were not influenced by the feed type, with animals fed either diet using the pen structures the same. Further, feeding times were the same suggesting the soybean-based protein diet was equally easy to eat and palatable as the animal-based protein diet. Behavior and feeding suggested soy-based diets do not alter time budgets or activities. There were no differences in the frequency or severity of pathologies for animals fed either diet. Respiratory (lung and trachea as a proxy to measure dust inhalation), gastrointestinal (small intestine as a proxy to measure digestive disturbances), and renal (kidney as a proxy for excretory stress) histopathology demonstrated neither diet was causing overt problems. One Welfare conclusions were feeding a soybean-based protein diet did not cause production or welfare issues. Further, soybean protein-based diets may be an environmentally sustainable alternative to currently used animal-based diets. Research examining different soybean protein concentrations and sources is warranted.

Corresponding author
Mark Flint, flint.71@osu.edu

## INTRODUCTION

Crocodiles and alligators are sourced from 30 countries for the legal trade of skins and meat. Approximately 1.5 M skins are sold annually around the world. Of these, commercial production of American alligators (*Alligator mississippiensis*) in the United States provides one third of the global market, increasing nearly 50% in the last decade. Demand for this species arises from it creating a high-quality leather product (*Medley, 1970*; *Nickum et al., 2017*; *Caldwell, 2021*).

For American alligator skins to remain competitive on a world stage it is important to optimize every aspect of the farming system including protecting the resource being sourced and optimizing the skin produced in a welfare-oriented and an environmentally sustainable way. In a modern global facility, this is achieved through supporting the species, addressing the health and well-being of the animals in the system, as well as minimizing the lifetime carbon footprint this production enterprise has on the environment and the people around it—from sourcing the eggs through to disposition of final product (*Center for Sustainable Systems (CSS), 2021*). Further, in our evolving understanding of conservation, environmentally and socially responsible businesses are driven to seek long term ecologically friendly production alternatives to mitigate negative practices that environmentally, culturally and socially benefit the local and global community through the practice of One Welfare (*Stephen & Wade, 2018*).

In the wild alligators have been recorded to feed on insects/arachnids, fishes, crustaceans, gastropods, mammals and other reptiles (*Coulson & Hernandez, 1983*; *Delany & Abercrombie, 1986*; *Hilevski, Cordero & Siroski, 2022*; *Rosenblatt et al., 2023*). It was historically thought that alligators were unable to digest plant-based proteins due to this species being specific "carnivores", with their limited enzymatic and insulin responses to carbohydrates (*Coulson & Hernandez, 1983*; *Hilevski, Cordero & Siroski, 2022*). Plant-based proteins have been adapted for commercial use in numerous studies and have shown to not negatively affect alligator health or growth rates (*Staton et al., 1990b*; *Reigh & Williams, 2013*; *DiGeronimo et al., 2017*; *Reigh & Williams, 2018*; *Hilevski, Cordero & Siroski, 2022*). Numerous studies have found that plant-based protein diets commercially produced for alligators and other aquaculture species were as digestible as animal based/ control diets (*Coulson & Hernandez, 1983*; *Staton et al., 1990a*, *1990b*; *DiGeronimo et al., 2017*; *Arriaga-Hernández et al., 2021*; *Hilevski & Siroski, 2021*; *Hilevski, Cordero & Siroski, 2022*). There are potential limitations using vegetarian diets in predominantly carnivorous species with a consideration being lack of other nutrients and amino acids when limiting feed to one source of protein (*Mariotti & Gardner, 2019*); however it appears from several studies alligators tolerate plant-based proteins well.

Unlike other regions of the world where soybean demand has caused deforestation (*Aragão et al., 2022*) and overuse of agrochemicals, soybean crops locally grown by U.S. farmers can potentially be managed in a sustainable, environmentally friendly enterprise that supports local communities (*U.S. Soy (USSOY), 2021*). Further, soybean meal—a

commonly used plant-based feed in many terrestrial animal feeds—has previously been demonstrated to show promise in aquatic species such as alligators (*Reigh & Williams, 2016*; *DiGeronimo et al., 2017*; *Reigh & Williams, 2018*; *Hilevski, Cordero & Siroski, 2022*). Consequently, it has a positive potential for use in the alligator production industry.

Soybean meal based feed has a protein content of up to 39% depending on genotype and serovar (*de Borja Reis et al., 2020*), which can be further concentrated post harvesting. These protein levels are sufficient to service the alligator feed industry. Commercially raised alligators in the United States are traditionally fed an animal-based protein derived pelleted diet with a protein concentration of up to 50%.

We assessed the impact of using two different feed types on the welfare, feeding behavior and growth characteristics of 2-year old American alligators housed in 13 replicated pens over the course of the final 8 months of conditioning leading up to harvest. This first-of-a-kind welfare-orientated feeding and nutrition trial examined if soybean-based protein commercial pelleted diet would influence growth, behavior, health and overall One Welfare when compared to traditional animal-based protein commercial pelleted diet.

## MATERIALS AND METHODS

### Experimental design

Using a randomized model, 1,728 alligators hatched on-farm from wild harvest eggs and held in a closed system in 2019 were fed one of two diets to determine if using a sustainable plant-based protein would negatively impact growth, gastrointestinal health and behavior.

Animals were housed across 13 communal pens (6.1 m × 5.5 m; seven soybean-based protein diets (SBP) and six animal-based protein diets (ABP)) with a stocking density average of 0.25 m$^2$ (0.21–0.29 m$^2$) per animal as per industry guidelines. A total of 928 animals were assigned to the soy treatment and 800 to the standard diet with mean starting weights of 12.6 kg (±2.8 kg SD) and 12.9 kg (±2.7 kg SD), respectively.

Animals were checked daily for signs of activity and general health and fed five times per week. All animals were housed in insulated fully enclosed pens to maintain a water temperature of above 25 °C and a high humidity. The pens were plastic lined on the floor and halfway up the sides to provide a minimum water depth of 40 cm. A total of 100% water changes occurred every 3 days. Each pen provided a dry area equaling 25% of the total pen surface area to allow choice of in or out of water. Natural low-level lighting was maintained.

No animals were prematurely removed from the trial. If an animal was found sick or injured, it was to be removed from the experiment and treated or euthanized based on independent veterinary recommendations as per the host farm's standard operating procedures.

All animals had a 12.5 mm Biomark® ATP12 microchip inserted in the right forelimb to allow identification. Based on the tag identification and treatment assignment described above, each individual in each pen and treatment was known to allow tracing throughout the entire trial.

Necropsies, performed at the start, middle and end of the trial were conducted to assess the gastrointestinal (mucosal erosion caused by acidity), renal (crystal formation caused by alkalinity) and respiratory (pneumonia secondary to feed dust inhalation) tracks for any possible negative impacts caused by eating a soybean-based diet over an extended period of time. Necropsies were performed by standard ventral midline entry by placing the animal on its back and making a single full length full skin thickness incision from the cranial aspect of the thorax to the cranial aspect of the pubis. Using a combination of sharp and blunt dissection of mesentery and connective tissue, soft tissues in the thorax and abdomen were exposed to allow direct access to the trachea, lungs, and intestine. The retroperitoneal kidneys were accessed by dissecting the dorso-caudal aspect of the abdomen. All tissues were collected by sharp dissection and immediately placed in 10% NBF labeled jars.

Animals were harvested for necropsy and all animals were harvested at the end of the trial as per the Humane Slaughter Guidelines, which requires spinal cord severance followed immediately by pithing (*AVMA, 2016*). In short, animals were electrostunned to induce loss of consciousness using a plate placed on the back of the neck powered by Smith-Root LR-24 Backpack Electrofisher (supplied by Smith-Root, Vancouver, WA, USA) delivering at 250 V at 30 Hz under a 30% duty cycle. The animal was then immediately transferred to a table where a trained operator severed the spinal cord using a 5 mm expanding to 12 mm thin-blade boning knife inserted between the skull and C1, and was pithed by inserting the boning knife blade into the brain case *via* the foramen magnum through the incision made during severing the spinal cord (*Flint et al., 2023*).

Both feeds were made by a commercial feed manufacturer to emulate the standard feed in all but protein source. Both feeds were nearly identical in composition for protein and fat percentages and identical for all minerals and elements. Crude protein was a minimum of 50%, crude fat a minimum of 12%, crude fiber a minimum of 4%, and phosphorous was maintained at 1% (Cargill™ 50% Gator). Feed volume was weighed to provide 5% bodyweight per week. At all times during the experiment, animals were fed more than they ate on a per pen basis.

Each pen was fitted with digital cameras for recording behavior and events for assessment of feed time duration, and activity levels. For all animals, behavioral assessments were performed at start, middle, and final 2-week periods of the trial for comparison.

All data were collected under The Ohio State University IACUC approval # 201800000096.

## Data
### Morphometric

On entry to the pen, all animals were tagged, belly width measured and weighed. Every 4 months, all alligators identified, reweighed and reassessed for health and well-being. At the end of the trial, all animals were measured and those animals which were harvested had their hides assessed for quality. To account for variances in starting size, the most informative measurement was the amount of change during the trial.

### Nutrition

For both of the food types used, the manufacturer maintained food composition analysis to determine quality and product consistency.

### Health

At the start, middle and end of the trial, necropsies per treatment were conducted to determine animal health in response to feed acidity or alkalinity and feed "dust". Histopathology was performed on gastrointestinal, renal and respiratory track samples and scored for any pathologies by an appropriately qualified veterinarian ranging from '0' for none to '3' for severe.

### Behavioral

Using a multi-track recorder, interactions were recorded for 2 weeks three times throughout the 8-month trial. We used multi-screen cameras to record continuously. Behaviors were recorded at every hour for 2 min and included interaction occurrence and movement (activity), feeding duration and the number of animals on platforms.

Group feeding duration was the amount of time spent with most of the group actively feeding. Total feeding duration was the amount of time spent with any animals observed feeding.

## Husbandry

All husbandry parameters were kept the same across treatments. Each food type was fed five times a week as a 100% ADFI diet based on animal numbers per pen and pen weights. Complete (100%) water changes were performed twice weekly. Natural industry lighting regimes were used.

## Analysis

All data was analyzed using descriptive statistics at the treatment level. Feed type/nutrition, behavior, stress, and resultant growth, hide condition and necropsy evaluation (pathology) was scored for multiple factor comparisons. Kruskal–Wallis chi-squared tests and ANOVA's as appropriate were performed to determine significance.

## RESULTS

### Mortalites

Between the start and the end of the trial a total of 33 animals were not Sp. Observed. It is assumed these animals either died or lost their PIT tags. Nine (1.1%) of those animals were from the ABP treatment and 24 (2.5%) were from the SBP treatment. This result was statistically significant however the biological significance is considered not important.

### Morphometrics

A total of 1,682 alligators were used to calculate changes in belly width between the start to the end of the trial. All animals and each treatment were non-normally distributed so

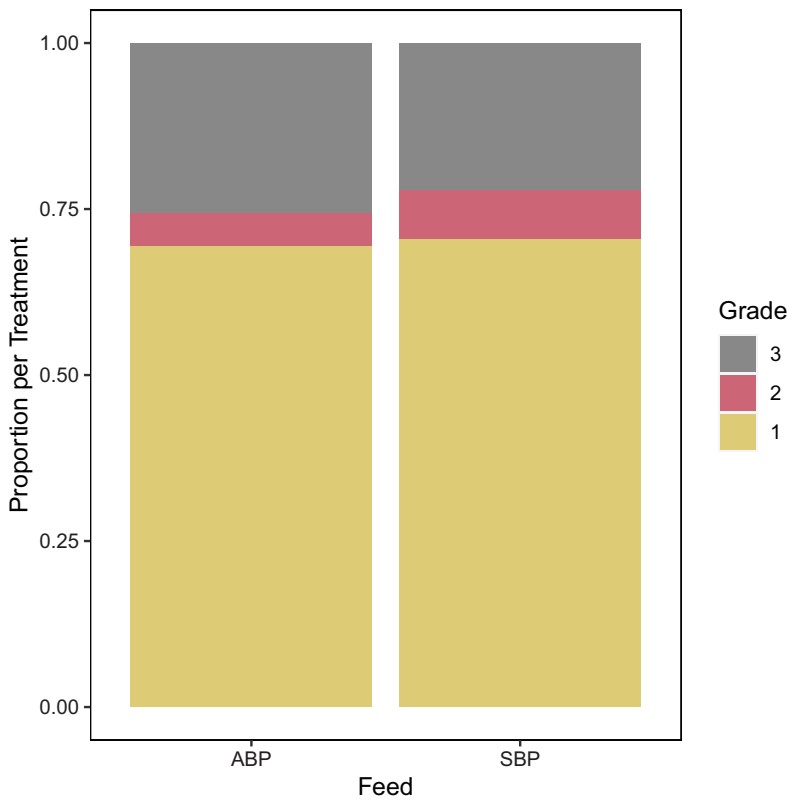

**Figure 1 Proportions of grades for Animal Based Protein (ABP) diets (*n* = 796) and Soybean Meal Based Protein (SBP) diets (*n* = 896) used in American alligators.**

non-parametric tests were used to analyse the data. The "Pen effect" was ignored for all morphometrics for this trial as it is assumed (based on authors' experience) that all alligator farming practices will result in pen effect due to the way animals are sorted to minimize aggressive interactions and it is an acceptable variance in all farm-based trials.

A total of 896 animals were fed SBP and tracked for change in belly width for the duration of the trial, while 786 alligators were fed ABP food and tracked for change in belly width. There was a significant difference between feed treatments ($p$ = 0.0009) with ABP animals growing more than SBP pens (5.9 ± 2.7 cm *vs.* 5.5 ± 2.4 cm).

A total of 1,677 alligators were used to calculate the changes in weight between the start and the end of the trial. Five animals did not have their weight recorded at the end of the trial. Most groups were non-normally distributed so both parametric and non-parametric tests were used to analyse the data. Pen effect was noted but ignored for this trial.

There were 893 alligators tracked for their change in weight in response to being fed SBP feeds and 784 tracked for their change in weight being fed ABP. There was no significant difference between treatments ($p$ = 0.84; ABP animals 11.0 (±3.9 kgs) and SBP animals 10.9 (±3.7 kgs)).

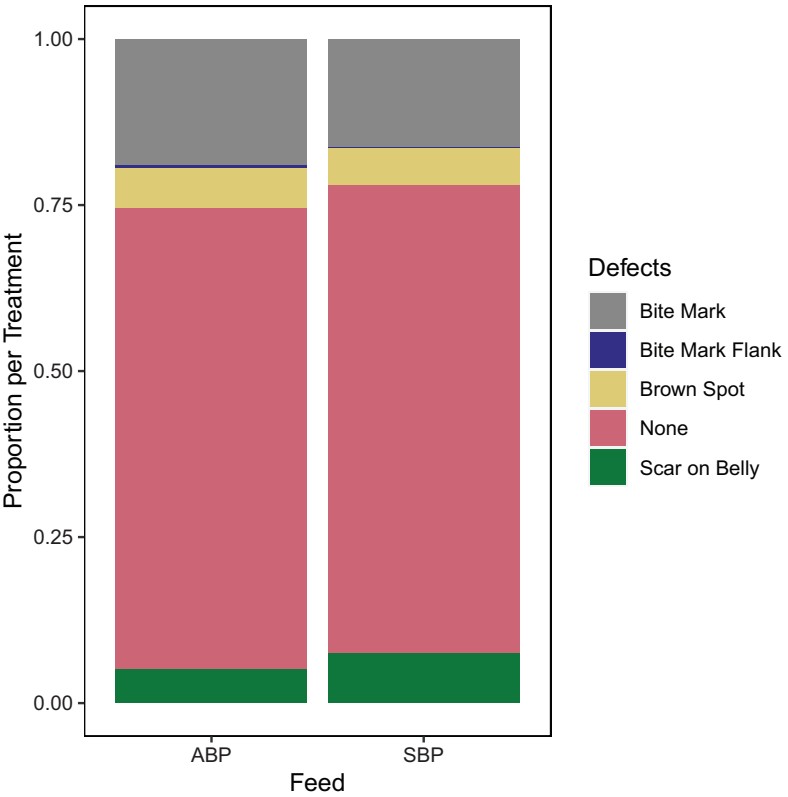

**Figure 2** Proportions of hide defects per diet showing similarity of bites, scars and brown spot on hides examined for animal based protein (ABP) diets ($n$ = 784) and soybean Meal based protein (SBP) diets ($n$ = 893) used in American alligators.

### Hide quality

A total of 719 hides were examined from the SBP and 616 were examined from the ABP. Of the SBP diet 70.4% were graded as Grade 1, 7.5% as Grade 2, and 22.1% as Grade 3 hides; while 69.3% were Grades 1 from the ABP diet, 5.0% as Grade 2 and 25.6% as Grade 3 hides (Fig. 1) ($p$ = 0.72, 0.08, and 0.15, respectively).

Further, breaking down to specific lesions of interest to the industry, there was no significant difference between diet for bite marks on the body ($p$ = 0.22) or flank ($p$ = 0.86); brown spot ($p$ = 0.73) or belly scars ($p$ = 0.08) (Fig. 2).

## Health (histopathology)

For animals across pens for each feed type, we compared the histopathology of (i) the gastrointestinal tract to see if the feed was having an impact on intestinal lining, (ii) the trachea and lungs to determine if feed was being aspirated or dusty; and (iii) the kidney to see if the diets were causing renal disturbance such as calculi or inflammation.

There were no noted differences in the proportion or presentation of pathologies for any of the examined tissues for either the SBP diet when compared with the ABP diet (Table 1).

**Table 1 Histopathology in American alligators fed a soy-based protein diet compared with an animal-based protein diet for a period of 8 months.**

| Tissue | Score* | Animal-based protein diet (ABP) (%) (n = 12) | Soybean-based protein diet (SBP) (%) (n = 14) | p-value |
|---|---|---|---|---|
| Small intestine | 0 | 66.67 | 71.43 | 0.45 |
| | 1 | 33.33 | 21.43 | 0.77 |
| | 2 | 0.0 | 7.14 | 0.41 |
| Kidney | 0 | 91.67 | 85.71 | 0.5 |
| | 1 | 8.33 | 14.29 | 0.5 |
| Lung | 0 | 33.00 | 14.29 | 0.26 |
| | 1 | 41.67 | 42.86 | 0.20 |
| | 2 | 25.00 | 42.86 | 0.24 |
| Trachea | 0 | 33.33 | 35.71 | 0.08 |
| | 1 | 33.33 | 35.71 | 0.08 |
| | 2 | 33.33 | 21.43 | 0.10 |
| | 3 | 7.14 | 7.14 | 0.41 |

**Note:**
* Where 0 is no detected pathology, 1 is mild anatomical changes such as sloughing, integrity change or inflammation to approximately 10% of the examined tissue field, 2 is moderate anatomic changes such as sloughing, integrity change or inflammation to over 20% of the examined tissue field; and 3 is severe anatomic changes such as sloughing, integrity change or inflammation to over 30% of the examined tissue field ± evidence of bacteria.

## Behavior

When looking at the total number of animals on platforms between the start and end of the trial there was a decrease for both diets (Fig. 3A).

Comparing the total number of animals on the platforms between SBP and ABP diets at the start of the trial, there were predominantly statistically significant differences; with fewer ABP animals on the platforms (Fig. 3A). However, by the end of the trial it flipped, there were predominantly statistically significant differences with fewer SBP animals on the platforms.

Comparison of the total number of activity actions observed at the start of the trial, there were few significant differences between diets. By the middle of the trial this had switched to ABP animals showing statistically less activity before maintaining this trend non-significantly by the end of the trial (Fig. 3B).

For SBP and ABP, the mean number of animals on the platforms and mean number of animals displaying activity for each hour block were mostly non-statistically significant differences in all sampling periods, with varying trends (Figs. 4A and 4B).

## Eating duration

The group time (when the most feeding was occurring) and the total feeding time (total time that individuals were observed feeding) was calculated. For all examined parameters, there was a pen effect (all $p \geq 0.34$). In general, eating duration was not influenced by feed type.

Comparing the time spent feeding between feed treatments at the end of the trial, there was no statistically significant difference for group feeding time ($p = 0.39$) or total group feeding time ($p = 0.52$).

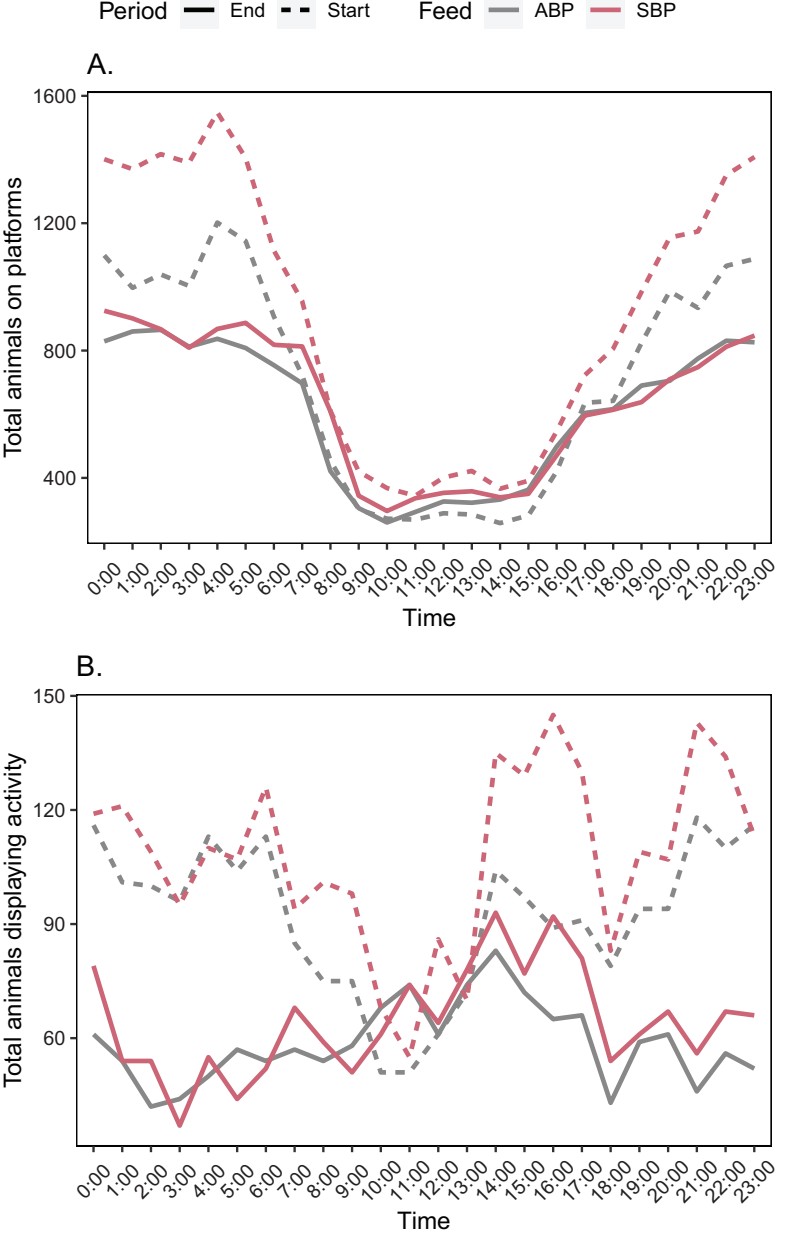

**Figure 3** Total number of American alligators (A) on a platform, (B) displaying some form of activity for each hour for animal based protein (ABP) diets and soybean meal based protein (SBP) diets at the start and end of the trial period.

## DISCUSSION

This first-of-a-kind welfare-orientated feeding and nutrition trial successfully determined that soybean-based protein commercial pelleted diet did not negatively influence liveweight gain, hide quality, behavior, health or overall One Welfare when compared to traditional animal-based protein commercial pelleted diet. It did negatively impact overall belly width gain.

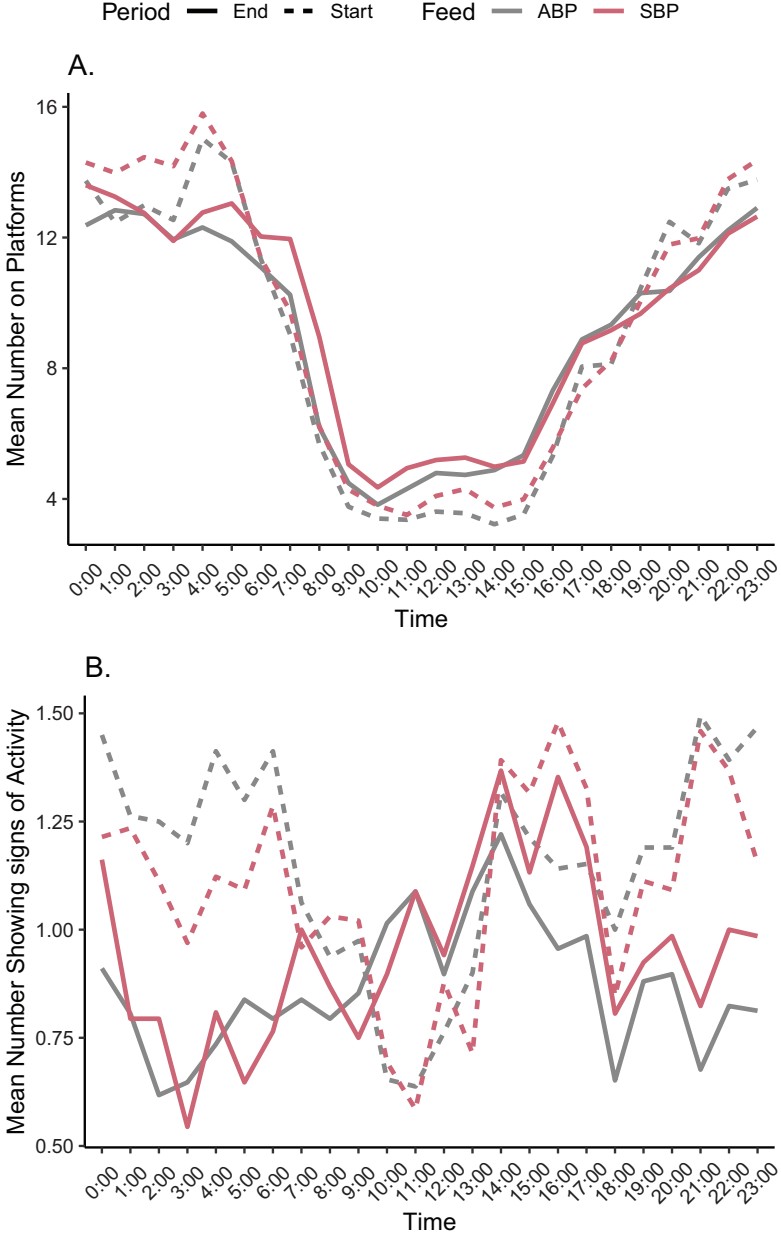

**Figure 4 Mean number of American alligators (A) on a platform, (B) displaying some form of activity for each hour for animal based protein (ABP) diets and soybean Meal based protein (SBP) diets at the start and end of the trial period.**

Alligators are used almost exclusively for their meat and hides meaning the size or the weight of the animal (as a proxy of yield of meat) and hide (measured as a belly width) are important parameters to optimize. Therefore, it is important to ensure any new diet being offered will not negatively impact either growth rate or the quality of the resultant skin or meat. Soy-based protein diet did result in a significant reduction of change in belly width (5.5 cm *vs*. 5.9 cm) when compared with ABP diet over an 8-month period. Soybean meal diets did not influence hide quality (based on independent qualified assessment at the

salted hide assessment stage). Further, it did not produce a change in weight, with alligators fed both diets growing an impressive average of 10.9 kg. This was similar to other studies examining growth rates of alligators fed plant based protein diets (*DiGeronimo et al., 2017*; *Reigh & Williams, 2018*; *Arriaga-Hernández et al., 2021*).

The pen effect may have influenced both the positive and negative findings (*St-Pierre, 2007*) and confirmatory studies are recommended.

Like all commercially managed species, good herd health is essential for optimal production. Small changes in high output systems can result in detrimental impacts. Nutrition is one of the key factors that can cause ill-health (*Adams, 2006*; *Britt et al., 2021*). Therefore, it was paramount to understand whether switching the protein source from animal-based to soy-based would have any negative effects on the health of the examined alligators. There were no noted differences in the proportion of pathologies for any of the examined tissues for the soy-based diet when compared with the ABP diet.

For animals across pens for each feed type, we compared the histopathology of (i) the small intestine to see if the feed was having an impact on intestinal lining; (ii) the trachea and lungs to determine if feed was being aspirated or causing an irritation; and (iii) the kidney to see if the diets were causing renal disturbance such as calculi or inflammation. For each of these tissues the proportions and presentation of pathologies were equal to that of the ABP diet being fed. Further, in most cases, the pathologies noted were less than a score of one (which was considered negligible to minor; Fig. 5) (*Flint et al., 2010*). These findings suggest being fed a diet of SBP feed for a period of 8 months will not negatively impact health of the alligators when compared to commercially available pelleted alligator feed.

Platform usage in alligator pens is speculated to be an important behavioral and physiological function where the animal exhibits a normal range of their behaviors by having choice to be in or out of water (*Manolis & Webb, 2016*). Further, it is potentially a required behavior to help with thermoregulation being able to get out of the water when too warm or cool. Diet did not influence platform use suggesting SBP was not altering behavior or physiology.

In most species held under human care, the time an animal spends feeding is an important welfare consideration. Too short a time and the time budget of behaviors can be skewed resulting in boredom and associated disruptive behaviors. Too long a time and the animal may tire or be out competed by cohorts before it is satiated. Throughout this experiment, we examined feeding as an amount of time for the group to consume most of the pellets fed out. Comparing the start, and end, and comparing SBP and ABP feed, there were no significant differences for period or feed. All feed was consumed in similar times; suggesting the soy-based diet was as palatable as the ABP diet and as easy to eat.

## Overall welfare conclusions

The Five Domains Model focuses on nutrition, environment, health, behavior and mental state, lending itself well to farm based studies being interpreted as a scale of overall welfare (*Mellor et al., 2020*).

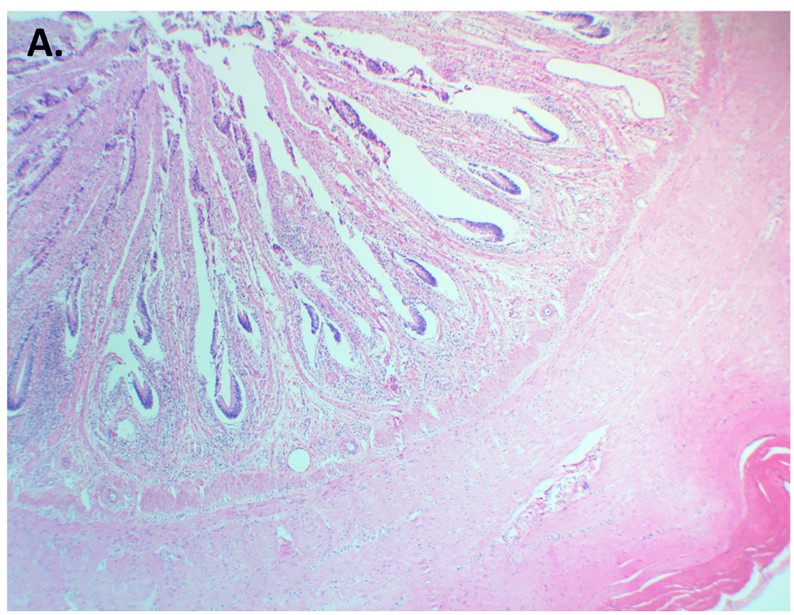

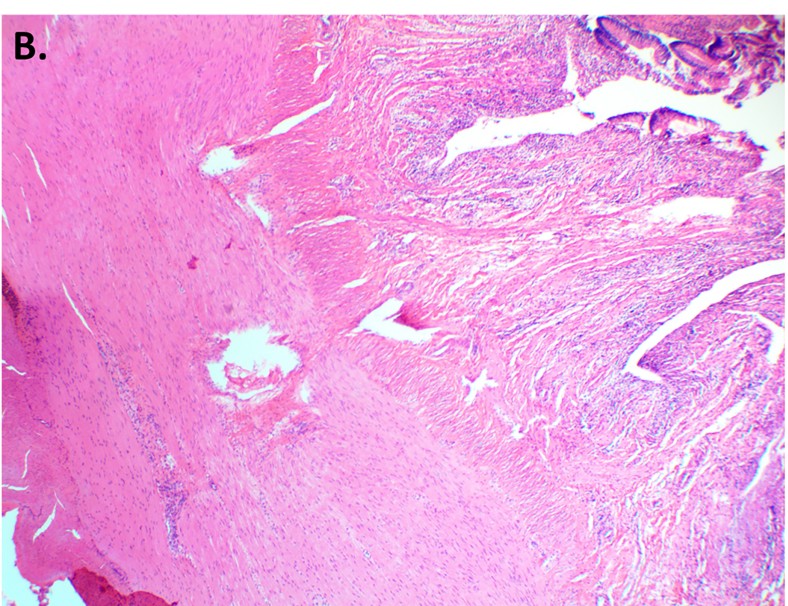

**Figure 5 Histological cross-section of small intestine (jejunum) for American alligators fed (A) Animal Based Protein diet or (B) Soybean Meal Based Protein diet.** Both animals scored 1 for minor inflammatory responses of the gastrointestinal tract. H&E. 10×.

For this research, the two types of nutrition, a 50% protein diet containing only soybean meal based protein and a 50% protein diet containing only animal based protein, we offered produced high quality products (skin and meat); the environment satisfied all of the animals' needs with no anomalies in environmental parameters, no excessive

pathologies, disease outbreaks or mortalities were detected; and behaviors were identical between feed types and did not appear to induce negative behaviors such as aggression.

In addition, the potential environmental benefits of feeding a sustainable renewable protein based resource produced locally within the United States that will help support primary producers and local communities has many positives.

Overall, we conclude that the welfare of the animals in this trial was good, and their mental state was all positive. Feeding a soybean-based protein diet was not detrimental when compared with feeding an animal-based protein diet. The overall One Welfare of this farming strategy was positive.

## ACKNOWLEDGEMENTS

The authors are grateful to Cypress Creek Farms LLC for the use of their facilities and animals to conduct this trial as well as their labor and expertise to farm the alligators and collect the data. We are also grateful to the commercial feed company who manufactured the feed for us.

### Funding

This work was supported by the Ohio Soybean Council (19-D-45). The funders had no role in study design, data collection and analysis, decision to publish, or preparation of the manuscript.

### Grant Disclosures

The following grant information was disclosed by the authors:
Ohio Soybean Council: 19-D-45.

### Competing Interests

The authors declare that they have no competing interests.

### Author Contributions

- Mark Flint conceived and designed the experiments, performed the experiments, analyzed the data, prepared figures and/or tables, authored or reviewed drafts of the article, and approved the final draft.
- Jaylene Flint conceived and designed the experiments, performed the experiments, analyzed the data, prepared figures and/or tables, authored or reviewed drafts of the article, and approved the final draft.

### Animal Ethics

The following information was supplied relating to ethical approvals (*i.e.*, approving body and any reference numbers):
The Ohio State University provided full approval for this study (2018A00000096).

## Data Availability
The raw data are available in the Supplemental File.

## Supplemental Information
Supplemental information for this article can be found online at http://dx.doi.org/10.7717/peerj.16321#supplemental-information.

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
