# Peer review of "Use of soybean as an alternative protein source for welfare-orientated production of American alligators (Alligator mississippiensis)"

_PeerJ, doi:10.7717/peerj.16321_

## Round 0.1 · original submission · Minor Revisions

Dear Authors

Please address the comments raised by the reviewers.

Reviewer 1 ·

Basic reporting

The manuscript entitled “Use of soybean as an alternative protein source for welfare-orientated production of American alligators” provides fundamental knowledge for future development of artificial diet for American alligators. However, the writing and presentation of data against publication in its current form. More details and considering comments are listed below.

Comments
1. Provide scientific name of this alligator in the title
2. “Soybean” or “soybean meal”?
3. Abstract should be merged into only one paragraph. Experimental design comprising dietary treatments, replication, and number of animals should be given. Detailed information such as proportion of soybean, type of animal-derived feedstuff, and proximate composition of the prepared diets, is essential and should be given in this part. Authors should specify “P-value” for any significant figures.
4. Introduction part did not make enough references to the latest literature on the subject. Background referring nutrient requirement, recent diet development for alligator, nutritional information of soybean, and the limitation to use in diet should be mentioned.
5. L64, clarify “soybean feed”

Experimental design

6. The amount of information given about the materials and methods is variable, with insufficient details being given about some of the methods and protocols, and there are a few points that need clarification: e.g. references for the methods on L124-128, methods for euthanasia and how to sacrifice and prepare the alligators before collecting the samples?
7. Ingredient proportion and proximate chemical composition of the two prepared diets should be summarized in Table. These details are very important since there was no any information about the levels of each feedstuff in diet formulation.
8. Number of alligators for each assessment (morphometrics, health, and behavior) should be given.

Validity of the findings

9. Since no significant differences were observed between ABP and SBP, the statement on L164-165 (“grew the more than”) was not supported by statistical analysis.
10. L167-169, grades 2 and 3 were not mentioned.
11. Table 1 was not mentioned on L176-181.
12. L184-198, correct capital/lowercase for subfigures in text, as well as on Figures 3 and 4
13. Discussion part did not make enough references to the latest literature on the subject. Remove headings in this part and separate the results by paragraph. The dietary levels of soybean in diets, and the benefits and limitation of using soybean in aquatic feed should be discussed.
14. Conclusion part should be able to stand alone, so that replace abbreviated term by full word. In addition, since the proportion of soybean in diet and type of animal-derived feedstuff are essential and may have significant effects on obtained results, these information should be included in Conclusion.
15. Table 1, replace treatment names from “Animal” and “Soybean” by “Soy-based protein diet or SBP” and “Animal-based protein diet or ABP”
16. Figures, each table should be able to stand alone, so that number of animals per assessment and y-axis legend should be given.

Additional comments

-

Reviewer 2 ·

Basic reporting

Basic reporting
Generally well written, clear and concise. Note a few minor typographical errors or grammatical errors as listed below. These should be corrected prior to publication

14 typo diets “have” not “has”
115 “were” identified
119 no hyphen between manufacturer and maintained
153 “trial” not “trail” typo
156 “with APB animals growing” not “grew” typo
209 “the size of the weight of the animal” is not clear. Size or weight perhaps?

Experimental design

Accepted. No further comment

Validity of the findings

No comment

Additional comments

No comment- article provides a great foundation for exploring alternative ethical food sources

Reviewer 3 ·

Basic reporting

No Comment

Experimental design

No Comment

Validity of the findings

No Comment

Additional comments

Introduction
- I suggest that the authors include a paragraph on American Alligator natural diets and what possible adjustments have been made for captive alligators.

Results
- 1728 alligators were recruited into the study but 1682 and 1677 were used for morphometric measures. The authors will need to account for the other crocodiles. If they were mortalities, which treatment groups did they come from?

Discussion
- Health (histopathology): The author should add pictures of the histology slides of the intestines. Although the difference was not statistically significant, what is the author's opinion of the SBP group having a higher score of 2 in the small intestines, and lung tissues on histopathology?

Annotated reviews are not available for download in order to protect the identity of reviewers who chose to remain anonymous.

---

## Round 0.2 · accepted · Accept

The authors have addressed all the comments and concerns raised by the reviewers.